# The Effects of Art Therapy on Anxiety and Distress for Korean–Ukrainian Refugee: Quasi-Experimental Design Study

**DOI:** 10.3390/healthcare11040466

**Published:** 2023-02-06

**Authors:** Soo-Yeon Kim, Jenny Seongryung Lee, Han Choi

**Affiliations:** 1Department of Medicine, Graduate School, Cha University, Seongnam 13488, Republic of Korea; 2Graduate School of Art Therapy, Cha University, Seongnam 13488, Republic of Korea

**Keywords:** Russia–Ukraine war, war refugees, mental health, anxiety, distress, art therapy

## Abstract

Since Russia invaded Ukraine in February 2022, there has been an urgent need to provide mental healthcare and share various practices for Ukrainian war refugees. This study urgently focuses on the need for art therapy to support the mental health of Ukrainian refugees, Koryo-saram, who are staying in the Republic of Korea due to the wartime emergency. It also examines the impact of art therapy intervention on anxiety and subjective stress. The single-session art therapy with the 54 Koryo-saram refugees aged 13-68 showed the effectiveness of the art therapy intervention. The results indicate that GAD-7 (*t* = 3.092, *p* = 0.003) and SUDs (*t* = 3.335, *p* = 0.002) were statistically significant within the intervention group. In addition, satisfaction assessments of the qualitatively analyzed participants showed that Ukrainian Koryo-saram had a positive experience of art therapy. Therefore single-session art therapy in this study demonstrated the efficacy of art therapy for the anxiety and subjective distress of Ukrainian Koryo-saram refugees. This result suggests that the intervention of art therapy as immediate mental healthcare for refugees facing war could benefit the mental health of Koryo-saram refugees.

## 1. Introduction

Since Russia invaded Ukraine on 24 February 2022, millions of people have been forcibly displaced. The war between the two countries has become one of the largest and most pressing humanitarian emergencies since the Second World War [1]. According to the UN refugee agency (UNHCR) data of 2022, the Russia–Ukraine War 22 (RUW-22) has caused many civilian casualties. It has destroyed civilian infrastructure, forcing people from their homes for safety, protection, and support. As a result, millions of refugees from Ukraine crossed the border into neighboring countries. Refugees are individuals who cannot or do not want to return to their country of origin, defined in the 1951 Refugee Convention as an essential legal document [2]. They must cross an international border to find safety in another country due to war, violence, conflict, or persecution; they often flee without property, a job, a home, or a loved one [2].

In total, 962 Ukrainian compatriots, known as Koryo-saram, entered the Republic of Korea (ROK) for safety, protection, and support in February 2022 [3]. Koryo-saram(s) are Koreans who immigrated to the former Soviet Union in the 1860s because of the anti-Japanese colonization movement and forced mobilization until Korean liberation from Japanese rule [4]. The Ministry of Justice of the ROK have currently issued visas for Koryo-saram and their families until the end of the RUW-22. However, only women, children, and older adults have fled from Ukraine and settled in the Koryo-saram communities, while their husbands and fathers are still in Ukraine because of conscription.

As of 28 February 2022, the ROK has permitted 3843 Koryo-Saram to stay longer with humanitarian assistance [3]. Those Koryo-saram, not directly facing the war, might have vicariously experienced trauma from news reports showing their neighborhood destroyed and witnessing on social media the suffering of their families and friends living in Ukraine. A study of university students in the neighboring Ukraine country the Czech Republic shows that their trauma is associated with a higher frequency of news monitoring and social media use. Particularly for females, the higher frequency of news and the use of social media have led to higher symptoms of anxiety and depression [5]. In addition, Ukrainian war refugees who have newly entered the ROK, and those who have spent extended periods in the ROK, also suffer from psychological anxiety and stress due to the loss of their homes and the separation from their families [6].

Experiencing war causes sudden tragedies without preparation, damaging their properties and other valuable assets, causing family separation and displacement, the death of a loved one, a lack of social support, and negative coping skills, adversely affecting civilians’ mental health [7]. They have to adapt to new cultures and languages while experiencing these psychological difficulties. For the reasons above, war affects individuals’ mental health and puts them at risk of severe psychological disturbances, including anxiety, depression, posttraumatic disorder (PTSD), and emotional difficulties [8,9].

The World Health Organization (WHO) promoted action on Ukrainian refugees’ urgent mental health needs and discussed refugees’ mental health needs, focusing on health and migration in March 2022 [10]. Experts initially focused on providing only the necessities for immediate assistance to arriving refugees but are now focused on the emergency provision of mental healthcare, sharing various best practices [10]. The reason is that children and adolescent refugees who have experienced war encounter many of the same struggles as their parents, such as separation from family, discontinued education, exposure to death and violence, inadequate nutrition, problems with acculturation, and discrimination [10]. Hence, regarding the mental health of refugees who have experienced war, anxiety and depression are the most common psychopathological symptoms [11,12,13]. Second, PTSD symptoms have also appeared [13,14,15], and third, somatic symptoms have also been reported [12,16]; fourth, female had higher stress levels in psychological and emotional states than men [17].

Studies on the mental health of war refugees have shown that their experience of war can cause, worsen, or exacerbate pre-existing mental health illnesses and, in some cases, cause significant distress [9,11,18,19,20]. Physical manifestations such as headache, unspecified pain or discomfort, and fatigue may arise among them, and cognitive errors may also occur [16]. According to the WHO, about one in five people will experience mental health problems in the next decade. One in ten people will suffer from severe illnesses such as posttraumatic stress disorders or mental illness [10]. People who experienced the war reported a greater risk of mental health complications, including anxiety and depression, compared to those who did not [8]. In a study that followed the mental health of war-injured refugees for eight years, the prevalence of psychiatric symptoms was elevated, levels of anxiety and depression were reported to be high, but relatively few people, however, received psychiatric therapy [21]. As such, many studies have reported mental problems caused by war, and rehabilitation programs and early psychological treatment with social support for their mental health are needed. However, there are few studies on mental health rehabilitation programs or early psychotherapy for refugees who have experienced war.

Art therapy specializes in a nonverbal psychological support methodology, which can safely express psychological discomfort and help individuals feel tranquility [22]. Many studies insist that art therapy is effective for people with verbal communication hardships and high psychological anxiety [23,24]. Images can express emotions and thoughts that cannot be explained in language alone [22]. Consequently, art therapy has often been applied to treat traumatized people, and good results have been reported in clinical practice. Art therapy has been perceived as incredibly helpful with diverse populations because art surpasses the walls of cross-cultural communication, for instance, distinctions in language, values, and notions of normality. In other words, it might adopt cultural values because images share a more ubiquitous meaning than words [25]. The therapeutic factors of adolescent and adult refugees who experienced war identified through art therapy research are as follows. First, art therapy is effective for anxiety, depression, trauma, and other general psychiatric symptoms found in war-affected states [26,27,28]. Artmaking can promote expression, support recovery through creative work, and help people stay in the present moment [22]. In addition, art therapy is mainly nonverbal expression and effectively reveals oneself without a language barrier [18]. In particular, verbal communication is difficult for refugees experiencing traumatic situations. Second, art media can provide an outlet to express negative emotions and help individuals visualize their trauma in a nonthreatening way. Because traumatic memories are encoded nonverbally [1,3], art therapy effectively treats trauma-related psychopathology [1,3,4]. Third, art therapy enables the expression of emotions and thoughts through art media and products [17]. Art production with both hands stimulates bodily sensations and provides rest, eventually reducing arousal [29].

As a result of the preliminary literature review, there were no records of art therapy studies on Ukrainian war refugees. However, some war-refugee-related papers were found when the scope was expanded to case studies of art therapy for refugees who survived war [30,31,32] and the effects of reducing anxiety, trauma, and stress [33,34]. In addition, a study was conducted on Korean immigrant adolescents [23,24]. According to these previous studies, it was reported that art therapy is meaningful for the emotional stability and trauma of war refugees and Koryo-saram immigrants.

The objective of this study is to provide urgent mental healthcare for Koryo-saram refugees entering the ROK and to assess the effect of art therapy on Ukrainian war refugees on anxiety and subjective units of distress. The composition of this study’s purpose is as follows: (a) The primary outcome evaluation: GAD-7 and SUDs, and (b) the secondary outcome evaluation: GAD-7 items, (c) the secondary outcome evaluation: satisfaction with the art therapy, (d) lastly, the secondary outcome evaluation: a qualitative analysis.

## 2. Materials and Methods

### 2.1. Design

This study used a one group pre- and post-test, quasi-experimental design. This study followed the Transparent Reporting of Evaluations with Nonrandomized Designs (TREND) reporting guidelines for quasi-experimental study designs [35].

### 2.2. Setting

Data were retrieved from July to September 2022 at three disclosed locations within the greater Seoul metropolitan area of the ROK.

### 2.3. Participants

In this study, the total number of participants was 54, between the ages of 13 and 68. Among these, 22 teenager and 24 adult participants were newly entered Koryo-saram war refugees. In addition, seven adults and one teenager who had stayed in the ROK for more than two years took part because they are also refugees, regardless of the length of their stay. According to the UNHCR, refugees are persons who are unable to return to their country of origin. These eight Koryo-saram were unable to return to Ukraine. All the participants were recruited via private organizations that immediately responded to the Ukrainian Koryo-saram war refugee welfare crisis. The inclusion criterion was as follows: (a) participants aged older than 13 years, (b) those who entered the ROK after the Russian invasion of Ukraine in 2022 or temporary visitors who cannot return to Ukraine, and (c) participants who willingly took part in the experiment. Exclusion criteria included: (a) taking psychotherapy since the arrival of the ROK, (b) having concurrent mental health conditions and undergoing psychotherapy or arts therapies since the arrival of the ROK, and (c) who did not agree to participate in the study.

#### Demographic Characteristics of Participants

The demographic characteristics of the participants are shown in Table 1. Of 54 participants in this study, 40.7% were adolescents, and 59.3% were adults. Of the participants, 72.2% were female, and 27.8% were male. All participants were Ukrainian and refugees from the RUW-22. Of their length of stay in the ROK, 7.4% were under one month, 70.4% were 1–5 months, 3.7% were 6–11 months, 14.8% were over 24 months, and 3.7% did not respond. Participants’ Korean language levels were 0% advanced, 9.2% were intermediate, 88.9% were basic, and 1.9% did not respond.

### 2.4. Assessments

Participants’ demographic characteristics, including age, gender, length of stay since arrival, and proficiency in the Korean language were collected. Before the experiment, a Korean-Ukrainian bilingual interpreter translated the Subjective Unit of Distress scale (SUDs) and satisfaction of art therapy.

#### 2.4.1. Generalized Anxiety Disorder-7 (GAD-7)

Generalized Anxiety Disorder-7 (GAD-7) is a self-administered 7-item anxiety screening tool to identify probable cases and measure the severity of GAD symptoms [35]. The GAD-7 items include (1) nervousness, (2) inability to stop worrying, (3) excessive worry, (4) restlessness, (5) difficulty in relaxing, (6) easy irritation, and (7) fear of something awful happening. In the assessment, participants are asked how often they have been bothered by anxiety-related symptoms over the past two weeks. Response options for each item range from 0 to 3 on a 4-point Likert scale (0 = not at all, 1 = several days, 2 = more than half the days, and 3 = nearly every day). Adding the scores of all seven items provides the GAD-7, and the total score ranges from 0 to 21. A score of 0–4 indicates minimal anxiety, a score of 5–9 is mild, a score of 10–14 is moderate, and a score of 15–21 is deemed severe. The GAD-7 has demonstrated good internal consistency, test–retest reliability, and convergent, construct, criterion, and factorial validity [36,37,38,39]. Cronbach’s alpha was 0.895 in this study, indicating solid internal uniformity. Pfizer Inc. holds the copyright, and the questionnaire is free to use. This study used the Russian version of the GAD-7, available on the Patient Health Questionnaire website [40].

#### 2.4.2. Subjective Unit of Distress Scale (SUDs)

Subjective Unit of Distress scale (SUDs) is a self-assessment tool to measure the intensity of distress or nervousness, distress, fear, anxiety, or discomfort on a scale of 0–100 from the absence of any distress (0) to discomfort is extreme (100) [41]. SUDs can be a personal assessment used by a mental healthcare provider to evaluate participants’ treatment progress and the success of their treatment plans. SUDs can be used regularly over the months of their treatment to gauge different areas of distress or disturbance [36]. Cronbach’s alpha was 0.85 in this study, indicating high internal consistency.

#### 2.4.3. Art Therapy Satisfaction

The Visual Analog Scale (VAS), a self-evaluation scale, assesses the overall satisfaction of participants [42]. A satisfaction questionnaire consists of six self-reporting questions to measure the participants’ overall satisfaction. Four are on a Likert scale, one is choosing “what most liked” in this study, and the last is written comments. The Likert scale is 1–5, with 1 indicating the lowest satisfaction level and 5 indicating the highest satisfaction level.

### 2.5. Procedure

The researchers conducted a single-session group art therapy session for 90 min for fifty-four participants in four groups. The evaluations to verify the effects of art therapy intervention (independent variable) on their generalized anxiety and subjective distress (dependent variable) were conducted before the beginning of the art therapy and immediately afterward for 20 min each. They also answered the demographic characteristics questionnaire at the beginning, immediately after signing the consent forms, and answered the satisfaction questionnaire after completing the art therapy. All participants’ data were documented in a case report from a professional art psychotherapist, and Ph.D. and MA candidates in art therapy performed all procedures. The sampling procedure is shown in Figure 1.

### 2.6. Intervention

Art therapy is an expressive method that facilitates participants to engage in artmaking and creative approaches in a safe circumstance [22]. This art therapy was conducted as an immediate response to the Koryo-saram refugees’ mental health. The art-based intervention consists of four stages: (1) exploring clay as an icebreaker, (2) creating safe places, (3) writing notes, and (4) sharing and creating a safe group place.

The researchers initially provided the carefully selected art medium, clay, the circular shape of the ground, and natural objects, based on various research for this population [24,25,26]. Clay involves intense tactile experiences and enables the creators to communicate and express their voices nonverbally [43]. Once they felt thoroughly relaxed and ready to move to the next stage, the art therapists introduced a safe place, making it either real or imaginary. Creating a safe space allows them to look ahead, maintain resources, and regain control [44]. Whether psychological or actual safe places, the Koryo-saram refugees could safely explore themselves by creating them. Then they shared feelings and emotional experiences related to using various materials at this stage. Afterward, participants wrote about their experiences in notes and shared their art with groups and therapists at this stage, creating positive relationships and mutual support. In the final stage, the participants placed each safe place on a large canvas for group work. The participants decorated the canvas altogether. Every participant’s safe place was harmoniously united on the canvas, and they became a tremendously safe group. In closing, they were given electric candles and simultaneously turned on the lights. New beginnings and possibilities were explored in preparation for the end of the sessions. Through these processes, participants created a visual representation of their responses to safe places with clay and natural objects, found their true feelings and emotions, and eventually regained self-identity and positive resources. Figure 2 shows some of the participants’ artwork.

### 2.7. Statistical Analyses

SPSS 23.0 was conducted for all statistical analyses (IBM Corporation, Armonk, New York, NY, USA). First, the paired sample *t*-test was conducted to determine the effect of art therapy on anxiety and distress. Second, to determine the pre- and post-effects of art therapy for each item of GAD-7, the paired sample *t*-test was conducted. Finally, a frequency analysis was conducted to determine satisfaction, and participant feedback through satisfaction was qualitatively analyzed. The level of statistical significance was set at *p* < 0.05.

### 2.8. Ethical Considerations

This study complied with the Declaration of Helsinki and was performed according to ethics committee approval. The researchers thoroughly explained the purposes, procedures, potential benefits and harms, and risks to participants, and their right to withdraw from the study at any time. The researchers collected informed consent, including photographic releases of their artwork, directly from adult participants. The researchers also collected teenager’s and their parents’ signatures as minor participants. The researchers also explained that participants’ information and responses remain confidential. No individuals were identified in any reports or publications resulting from this study.

## 3. Results

### 3.1. Primary Outcome Evaluation: GAD-7, SUDs

The paired sample *t*-test was performed to test the difference between the pre- and post-test measurements within the intervention group, as shown in Table 2. The GAD-7 (*t* = 3.092, *p* = 0.003) and SUDs (*t* = 3.335, *p* = 0.002) were statistically significant within the intervention group. In addition, an independent *t*-test was conducted for the GAD-7 and SUDs evaluation to determine the differences between groups by age and gender. However, there was no significant difference in results between the groups.

### 3.2. Secondary Outcome Evaluation: GAD-7 Items

The paired sample *t*-test was performed to test the differences between the pre- and post-tests of GAD-7 within the intervention group, as shown in Table 3. The change in items of GAD-7: inability to stop worrying (*t* = 3.369, *p* = 0.001), excessive worry (*t* = 3.471, *p* = 0.001), restlessness (*t* = 2.207, *p*= 0.032), easy irritation (*t* = 2.704, *p* = 0.009), and fear of something awful happening (*t* = 2.195, *p* = 0.033) were statistically significant within the intervention group.

### 3.3. Secondary Outcome Evaluation: Satisfaction with Art Therapy

The result of the satisfaction with art therapy is shown in Table 4. In total, 88.8% of the participants agreed that art therapy was interesting. A total of 86.8% of the participants found that art therapy made them feel safe. A total of 87.0% of the participants felt that 90 min of art therapy was enough. A total of 77.7% of the participants expressed that they would take part in art therapy again. Lastly, 54.7% of participants found that the setting and the environment, creating art, and discussing the artwork made during the art therapy session was enjoyable aspects of art therapy.

### 3.4. Secondary Outcome Evaluation: A Qualitative Analysis

The qualitative analysis was conducted via the satisfaction evaluation with the participants’ experience of art therapy is shown in Table 5. The participants recreated their houses and their pets with clay and made art related to their husbands and fathers, who they could not be with. In doing so, they shared their concerns and worries and shed tears or expressed their anger freely as a part of their work. Through that process, they consoled and supported one another. As a consequence, four categories were derived: “Expression of positive emotions”, “Experience of a sense of security”, “Self-emotional awareness”, and “Plans”. Based on these four, the central theme, “Relief of anxiety through positive emotional strengthening”, was generated.

## 4. Discussion

This study aimed to urgently conduct art therapy for mental health management for Koryo-saram refugees staying in the Republic of Korea (ROK) due to the Russia–Ukraine war and to investigate the effect of art therapy intervention on anxiety and subjective pain. To this end, the levels of the GAD-7 and SUDs and art therapy satisfaction were evaluated for 54 participants, and the satisfaction feedback of the participants was qualitatively analyzed.

First, it is possible to find that art therapy in a single session effectively reduces participants’ anxiety and distress. It was confirmed that all participants in the art therapy had lower results than the pre-evaluation. This is consistent with the therapeutic factors that art therapy is effective for anxiety, depression, trauma, and other common psychiatric symptoms [26,27,28], and previous art therapy studies [1,3] showed similar results. Furthermore, war refugees may experience increased anxiety due to difficulties in language communication, and the assertion that this art therapy can reduce their stress is supported [23,24]. The similarities between this study and previous studies are the age distribution, group art therapy, and duration of the art therapy intervention, approximately 90 min. The differences are the nationality of the participants, the length of stay, the number of art therapy sessions, and the program’s content. For these reasons, the findings of this study need to be interpreted carefully. However, this study can be used as primary data because art therapy was undertaken urgently to care for the mental health of refugees in war situations.

Second, while art therapy was effective for the Koryo-saram war refugees, there was no significant difference in anxiety and subjective pain level depending on age and gender. This is because the art therapy program did not proceed with age and gender differences. Mental pain caused by war can lead to severe psychological disorders, including anxiety, depression, PTSD, and emotional difficulties [6,8,9]. Even Koryo-saram refugees who have not experienced the war directly could experience vicarious trauma just by coming into contact with it through the news or social media [5]. However, it is believed that the art therapy conducted for this study did not reveal any differences between the groups because it aimed at caring for current painful emotions and experiencing psychological stability by creating an image of one’s own safe space, regardless of age or gender.

Third, examining the pre–post-therapy changes in GAD-7, all seven items were lowered after art therapy. Among the seven items, five of them: “Inability to stop worrying”, “Excessive worry”, “Restlessness”, “Easy irritation”, and “Fear of something awful happening” showed significant results. Due to the causes of the sudden tragedy of war, such as separation without preparation, damage to property and other assets, and negative coping skills due to a lack of social support [7], war refugees experience psychological distress. Moreover, they might not be consciously aware of their psychological distress in these circumstances, but the anxiety and stress can manifest as somatic discomfort [10]. Traumatic memories also are encoded nonverbally [1,3]; therefore, expressing trauma-related memories through images is helpful for treatment [1,3,4]. Art therapy is one of the nonverbal psychotherapeutic techniques that can help people to safely express their psychological discomfort and help them to feel comfortable, which is often difficult to express verbally alone [22]. For those who have experienced the shocking situation of war, using both hands to touch an art medium and create artistic outcomes stimulates the bodily senses and relaxation [29]. From the experiences the participants had, it can indicate that they would have experienced mental and physical relaxation, which eventually lowered their anxiety. However, some participants’ anxiety levels were shown to be increased for some items. From much art therapy research, some artmaking therapeutic aspects are experiencing traumatic situations, expressing negative emotions through art media, visualizing their trauma as nonthreatening, and recognizing current emotions [23,24]. From the research above, the participants’ anxiety may have been expressed through negative emotions that they had not been aware of, which might have increased their anxiety temporarily. This also combines with the anxious feelings from experiencing war, an uncertain future, family separation, and traumatic experiences [11,12,13,16]. Therefore, it suggests that art therapy was significant for the participants.

Fourth, by analyzing the qualitative satisfaction of the level of experience of art therapy and the comments of the participants, the war refugees of Koryo-saram had a positive experience of art therapy. They also experienced a sense of safety within the groups. Their feedback was consistent with the experience, significance, and therapeutic factors of one of the studies on young Korean immigrants [24]. The participants rated “yes” on the need for art therapy as a part of their satisfaction evaluation, which means they wanted to receive it again. Despite the language barrier, researchers could communicate with participants via written comments and artwork. As a result, researchers could hear their voices and thoughts.

### 4.1. Strengths

This is the first study of mental health assessment and early psychotherapy for war refugees. It has been suggested that art therapy is effective as a method of urgent treatment in crises and the results can be used as primary data for future research.

### 4.2. Limitations

There are several limitations to this study. Due to urgent psychological support and participants’ unstable schedules, it was conducted in a single session. Furthermore, there was no control group or screening stage. Moreover, the small sample size was too small to generalize the study results. Consequently, a follow-up study on participants’ mental health status is necessary to increase the evidence. However, there is a limitation in this follow-up because refugees are in the resettlement process, which makes it hard to contact them in the long term.

### 4.3. Implications

This study found that immediate mental healthcare for refugees exposed to war and trauma is essential. Overall, the mental condition of refugees residing in the ROK was normal. However, participants’ anxiety levels, which before art therapy were marked as high based on GAD-7, decreased after art therapy. Future research suggests comparing the mental health status of war refugees who have received immediate mental healthcare and those who have not received any. We are also proposing a study that looks at the state of refugees who have received mental healthcare after three months of follow-up.

## 5. Conclusions

The single-session art therapy in this study showed the efficacy of the art therapy intervention for the anxiety and subjective distress of the Koryo-saram refugees who experienced the war. In the unique situation of war and the urgent need for mental healthcare, this study revealed that art therapy has turned into a freeing experience. This allows participants to release their experiences of pain safely and can reduce anxiety. Furthermore, the need for immediate psychological and emotional support for these populations has been confirmed. Art therapy, a nonverbal orientation method, is effective for individuals who have experienced trauma because the trauma is encoded as nonverbal memories in the body and brain. Again, the anxiety of most participants based on GAD-7 was not high, and it was a one-time therapeutic approach; careful interpretation will be necessary. However, it is hoped that this study can serve as primary data for future research into the early psychotherapeutic approach targeting war refugees.

## Figures and Tables

**Figure 1 healthcare-11-00466-f001:**
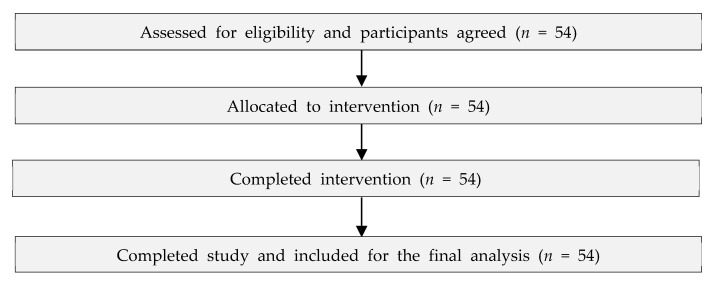
The sampling procedure.

**Figure 2 healthcare-11-00466-f002:**
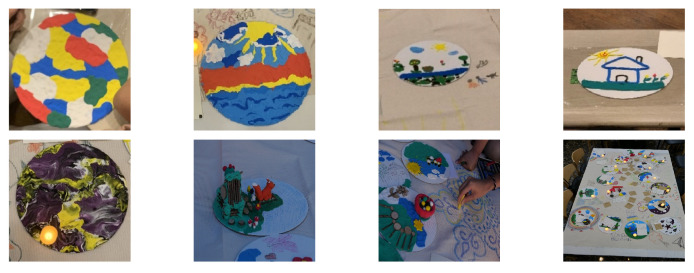
Participants’ artwork.

**Table 1 healthcare-11-00466-t001:** Demographic characteristics of participants.

Variables	Sub-Variables	Number	Percentage (%)
Age (years)	Teenager (13–18 years)	22	40.7
	Adult (19–68 years)	32	59.3
Gender	Male	15	27.8
	Female	39	72.2
Country of origin	Ukraine	54	100
Length of stay since arrival (months)	<1	4	7.4
	1–5	38	70.4
	6–11	2	3.7
	>24	8	14.8
	No response	2	3.7
Korean language level	Advance	0	0
	Intermediate	5	9.2
	Basic	48	88.9
	No response	1	1.9

**Table 2 healthcare-11-00466-t002:** Comparison of measurements within the intervention group.

Variables	Pre	Post	*t*	*p*
	M ^a^ (SD ^b^)	M ^a^ (SD ^b^)		
GAD-7 ^c^	4.61 (4.55)	3.29 (4.61)	3.092 **	0.003
SUDs ^d^	33.70 (22.92)	29.25 (23.83)	3.335 **	0.002

^a^ Mean, ^b^ standard deviation, ^c^ General Anxiety Disorder-7, ^d^ Subjective Unit of Distress scale, ** significant at *p* < 0.01.

**Table 3 healthcare-11-00466-t003:** Comparison of detailed analysis of GAD-7 within the intervention group.

Items of GAD-7	Pre	Post	*t*	*p*
	M ^a^ (SD ^b^)	M ^a^ (SD ^b^)		
1. Nervousness	0.85(0.88)	0.69(0.87)	1.589	0.118
2. Inability to stop worrying	0.72(0.90)	0.35(0.62)	3.369 ***	0.001
3. Excessive worry	0.85(0.86)	0.48(0.69)	3.471 ***	0.001
4. Restlessness	0.69(0.72)	0.50(0.77)	2.207 *	0.032
5. Difficulty in relaxing	0.43(0.77)	0.33(0.70)	1.093	0.279
6. Easy irritation	0.70(0.86)	0.48(0.81)	2.704 **	0.009
7. Fear of something awful happening	0.69(0.99)	0.46(0.84)	2.195 *	0.033

^a^ Mean, ^b^ standard deviation, * significant at *p* < 0.05, ** significant at *p* < 0.01, *** significant at *p* < 0.001.

**Table 4 healthcare-11-00466-t004:** Satisfaction of art therapy.

Questionnaire	Range	Frequency (*n*)	Percentage (%)
Q1. Did you find art therapy interesting?	Yes	16	29.6
	Quite yes	32	59.2
	Neutral	4	7.4
	Not quite	2	3.7
	Not at all	0	0
Q2. Did art therapy make you feel safe?	Yes	18	34.0
	Quite yes	28	52.8
	Neutral	4	7.5
	Not quite	2	3.8
	Not at all	1	1.9
Q3. Was there enough time for art therapy?	Yes	15	27.8
	Quite yes	32	59.2
	Neutral	4	7.4
	Not quite	2	3.7
	Not at all	1	1.8
Q4. Do you want to do art therapy again in the future?	Yes	5	9.2
	Quite yes	37	68.5
	Neutral	7	13.0
	Not quite	3	5.6
	Not at all	2	3.7
Q5. What aspects of art therapy did you enjoy the most?	Setting and Environment	9	17.0
	Creating art	7	13.2
	Discussion about artwork	8	15.1
	All three of above	29	54.7

**Table 5 healthcare-11-00466-t005:** Qualitative analysis satisfaction of art therapy.

Experience of Art Therapy by Participants	Category	Theme
I appreciated and felt better while receiving the art therapy.	Expression of positive emotions	Anxiety relief through positive emotion reinforcement
I enjoyed all the activities within the art therapy process.		
It was fun to play with clay and draw pictures on fabric.		
I realized we were together through this program, so I am not alone.	Experience a sense of security	
Thank you for being able to escape from constant worry and fear.		
I felt safe and good because of the comfortable and positive atmosphere and place.		
I could take care of my feelings and understand myself better.		
I felt safe and comfortable in this art therapy session.		
I liked everything because it felt like I was being understood how I felt.	Self-emotional awareness	
I could feel what the Ukrainians felt.		
May peace be brought to Ukraine.	Hopeful future	

## Data Availability

The data supporting the findings of this study are available from the corresponding author upon reasonable request.

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
