# Peer review of "The Effects of Art Therapy on Anxiety and Distress for Korean–Ukrainian Refugee: Quasi-Experimental Design Study"

_healthcare, 2023, doi:10.3390/healthcare11040466_

Round 1
Reviewer 1 Report
This is a timely and important issue to explore that I believe will merit publication. The aims of this study were to examine the impact of art therapy intervention on anxiety and subjective stress as a result of art therapy intervention.
There is, however, issues that must be resolved before the study can be accepted for publication.
1. Introduction
· Whether the goal of the study may be to goal of this study is to provide urgent mental health care?
· Lines 123-127: „Koryo-saram refugees may undergo high anxiety and subjective discomfort due to the psychological distress driven by war and the anxiety of adjusting to a new culture, language difficulties, and uncertain futures; art therapy would reduce anxiety and traumatic distress.” - Please present it in the form of a hypothesis.
· Lines 127-129: „The primary purpose was to provide immediate art therapy, and the secondary objective was to evaluate anxiety and subjective distress. The final was the satisfaction evaluation with the art therapy intervention.” - Unnecessary repetition of information.
2.3. Participants
· Provide inclusion and exclusion criteria.
· Lines 142-143: „Also, seven adults and one adolescent participant, who are not refugees, staying in Korea for over two years, participated in this research.” - Why were non-refugees included in the study?
2.4. Assessments
· Have the questionnaires used been validated?
· 2.4.1. Generalized Anxiety Disorder-7 (GAD-7) – Please add Cronbach's alpha. Did you need permission to use the questionnaire?
· 2.4.2. Subjective Unit of Distress Scale (SUDs) – Please add Cronbach's alpha. Did you need permission to use the questionnaire?
2.5. Procedure
· Lines 210-211: ”The researchers conducted a single-session group art therapy at three locations in 210 South Korea from July to September 2022.” - Unnecessary repetition of information.
2.6. Intervention
· Lines 224-225: „It was a one-day, two hours of art-based intervention” - Unnecessary repetition of information.
· Line 229: „based on various research for this population” – Please add references.
4. Discussion
· Lines 452-562: „In the participant satisfaction survey, the question "Was this art therapy interesting?" 59% were Quite Yes and 30% great. For the following question, "Did you feel safe during art therapy?" 54% was Quite Yes. 33% Great. For "Did you feel safe during art therapy?" Quite Yes was the most frequent at 59%, followed by Great at 28%. For "Was there enough time for the art therapy?" Quite Yes was the most frequent at 59%, followed by Great at 28% and "Do you want to do art therapy again in the future?" Quite Yes was the most frequent with 69%, followed by Great 9%. For the question, "What part of art therapy did you enjoy the most during the program?" 54% answered that all three, art therapy setting and environment, discussion about artwork, and creating art, followed by the art therapy setting and environment 17%, discussion about artwork 15%, creating art 13%, and other opinions 2%.” - this is information from the results section.
5. Conclusions
· Line 497: „[43]” - do not include references in conclusions.
Reviewer 2 Report
The authors develop a study in which a very interesting and topical subject is shown. It could even be said that it is novel and innovative since there is not much information in the scientific literature compared to other more common topics within the same family of concepts and descriptors.
The manuscript entitled "
The Effects of Art Therapy on The Anxiety and Subjective Distress for Ukrainian War Refugees "Koryo-saram": One Group Qua-si-Experimental Design" presents a number of features that invite the reader to show interest in the subject matter. In any case, a number of considerations and observations follow:
- The title is too long. It gives a lot of information but we should try to summarize without losing the main idea.
- The abstract has a logical structure. Although it ends by explaining the results, it would be interesting to put some more concrete data and end with a sentence referring to the conclusion. It would also be interesting to include more information about the sample, such as the percentage of women or men.
- The keywords are appropriate. If they are all of equal importance, alphabetical order should be used to show the keyword relationship.
- The general objective in the last paragraph of the introduction should also be broken down into specific objectives, appropriately numbered. These specific objectives should guide the presentation of the Results -in that particular section-. Likewise, hypotheses could be put forward (and also listed) so that later, in the discussion, they could be confirmed or not.
- If the information was obtained in September 2022, was the reported version of SPSS used, perhaps a more advanced one such as SPSS version 28?
- The first table of sociodemographic results regarding sex, age, etc. should be located in the Participants, within the Method.
- The results are adequate although they should be structured to a series of specific objectives.
- The discussion is interesting and allows the information obtained to be contrasted with other publications seen previously.
- It would be appropriate to expand the number of references from the last five years.
Thank you very much for your attention.
Reviewer 3 Report
Review of
The Effects of Art Therapy on The Anxiety and Subjective Distress for Ukrainian War Refugees "Koryo-saram": One Group Quasi-Experimental Design
The study explores the effectiveness of art therapy in reducing anxiety and subjective distress of war refugees. This particular study focuses on ethnic Koreans who fled Ukraine after the Russian invasion and sought refuge in Korea. I have the following comments for the authors:
1. The manuscript needs extensive language editing.
2. The effectiveness of art therapy in general needs to be more thoroughly discussed in the introduction. Detailing contexts when it is effective, reasons why it is effective and mechanisms through which it is effective. Only then relate art therapy to the context of helping refugees.
3. The sample of the study is unfit to test whether art therapy is effective for war refugees because of several reasons:
a. The lack of a control group;
b. The low sample size;
c. The heterogeneity of the sample;
d. The sample is comprised partly of individuals who are not refuges.
4. The assessment procedure is not sufficient to determine whether art therapy helped at all. Using the same scales to assess the same person just a few hours after an activity will produce some effect, regardless of what the activity was. Therefore, no real conclusions can be drawn from the data as the study is inherently flawed in this regard.
5. The sample is completely underpowered to run an ANCOVA.
6. The amount of multiple comparisons conducted in the manuscript drastically increases the likelihood of type 1 errors.
7. The qualitative part of the study is underdeveloped and can seem as an afterthought, it does not really add anything useful to the study because of how it is presented. If the qualitative part would have been more fully developed, perhaps it would be more useful, but that would require repeating the interviews.
8. No robust conclusions can be drawn; therefore the discussion section is mostly speculation.
I am sorry to say this, but overall, the article has many fundamental flaws and I see no real way of improving it except by conducting a different study following the best practices for experimental research. I hope this review does not discourage the authors from continuing their work. I wish them the best of luck in their ongoing and future research.
Reviewer 4 Report
The authors of the paper decided to analyze the use of art therapy to support the mental health of Ukrainian refugees who are staying in the Republic of Korea due to a wartime emergency. They also examine the impact of art therapy intervention on anxiety and subjective stress as a result of art therapy intervention.
The research objectives (the primary one and final one) were formulated correctly. No hypotheses were formulated, but there is no need for such research. Moreover, the small sample size was small to generalize the study results which was emphasized by the authors in the ‘Limitations’ section.
Authors’ English is correct. The paper requires some editorial/technical amendments but not many linguistic corrections.
The literature presented in the references is relevant and up-to-date. There are some editorial errors or omissions and appropriate complements are required:
– In case of the source no. 9, the full volume (issue) number is: 9(5).
– In case of the source no. 10, the article number 29 should be included instead of the page range 1-41.
– In case of the source no. 10, the correct DOI number is: https://doi.org/10.1186/s12914-015-0064-9.
– In case of the source no. 23, the correct volume (issue) number is: 27(3).
– In case of the source no. 23, the page range should be: 121-129.
– In case of the source no. 28, the correct order of the authors should be: Spiegel, D.; Malchiodi, C.; Backos, A.; Collie, K.
– In case of the source no. 35, the full page range is: 1092-1097.
– In case of the source no. 35, the unnecessary word ‘lamp’ is included among the authors’ names.
– In case of the source no. 37, the full page range is: 317-325.
The results are presented in a clear way and the discussion conducted correctly.
Reviewer 5 Report
Although the study is One Group Quasi-Experimental Design, why was a control group not sought?
There is a gender bias, since women predominated in a percentage of 72%. Do you think that the appreciation of art in terms of gender is different in terms of gender? in this study it is not clear.
Do you think that the lack of knowledge of the Korean language could have influenced the appreciation of art? 89% of the participants only had a basic knowledge of the language and this could be a further limitation of the study.
Round 2
Reviewer 3 Report
Review of the revised manuscript
The Effects of Art Therapy on Anxiety and Distress for Korean-2 Ukrainian Refugee: Quasi-Experimental Design Study
The revised manuscript, while somewhat addresses the comments I made in my previous review, still is not suitable for scientific publishing. Consider rewriting this article as a press release or as a communication to stakeholders and interested parties, but not as a research article. The article, unfortunately, does not meet the criteria for a scientific publication because of its methodological shortcomings which fundamentally cannot be addressed through any revision.
I wish them the best of luck in their ongoing and future research.
Author Response
Point 1: The revised manuscript, while somewhat addresses the comments I made in my previous review, still is not suitable for scientific publishing. Consider rewriting this article as a press release or as a communication to stakeholders and interested parties, but not as a research article. The article, unfortunately, does not meet the criteria for a scientific publication because of its methodological shortcomings which fundamentally cannot be addressed through any revision.
Response 1: Thank you for taking the time to read and for suggestions on our study. It really has been helpful to our study.
Reviewer 5 Report
The article can be accepted once the review is done.
Author Response
Thank you for taking the time to read and for suggestions on our study. It really has been helpful to our study.